# A Robust Brain MRI Segmentation and Bias Field Correction Method Integrating Local Contextual Information into a Clustering Model

**Zhe Zhang** [1] and **Jianhua Song** [1,2,*]

1   Electronic Engineering College, Heilongjiang University, Harbin 150080, China; 2171313@s.hlju.edu.cn
2   College of Physics and Information Engineering, Minnan Normal University, Zhangzhou 363000, China
*   Correspondence: 98dg_sjh@163.com; Tel.: +86-132-5160-5198



**Featured Application: The results of brain MRI segmentation can be used to extract the tissues or regions of interest, and even help doctors to determine the location of the diseased tissue.**

**Abstract:** The segmentation results of brain magnetic resonance imaging (MRI) have important guiding significance for subsequent clinical diagnosis and treatment. However, brain MRI segmentation is a complex and challenging problem due to the inevitable noise or intensity inhomogeneity. A novel robust clustering with local contextual information (RC_LCI) model was used in this study which accurately segmented brain MRI corrupted by noise and intensity inhomogeneity. For pixels in the neighborhood of the central pixel, a weighting scheme combining local contextual information was used to generate the corresponding anisotropic weight to update the current central pixel and thus remove noisy pixels. Then, a multiplicative framework consisting of the product of a real image and a bias field could effectively segment brain MRI and estimate the bias field. Further, a linear combination of basis functions was introduced to guarantee the bias field properties. Therefore, compared with state-of-the-art models, the segmentation result obtained by RC_LCI was increased by $0.195 \pm 0.125$ in terms of the Jaccard similarity coefficient. Both visual experimental results and quantitative evaluation demonstrate the superiority of RC_LCI on real and synthetic images.

**Keywords:** image segmentation; brain MRI; intensity inhomogeneity; bias field correction; weighting scheme

## 1. Introduction

Magnetic resonance (MR) image segmentation is a key step after magnetic resonance imaging (MRI), and its results directly affect diagnosis and treatment [1–3]. Numerous models have been proposed to achieve this, including clustering models [4–6], level set models [7–10], active contour models [11,12], and so on [13–15]. Accurate segmentation results have important guiding significance for clinical diagnosis. However, MRI quality can be easily destroyed by noise and intensity inhomogeneity because of a series of complex reasons [16]. Intensity inhomogeneity, which is also called bias field, appears as the smooth and slow change of the gray value of a pixel for the same tissue [17]. Hence, these defects in the process of brain MRI segmentation directly lead to segmentation errors [18].

Among all the segmentation methods, the clustering model is widely used, for which hard and soft clustering are the two main methods [19]. K-means clustering is the most representative model of hard clustering. Soft clustering can be further subdivided into mixture models and fuzzy c-means (FCM) [20], where FCM assumes pixels pertain to more than one class and has been widely

studied and applied. However, it is worth noting that FCM does not take any spatial information into consideration [21].

Many modified models incorporate spatial information into FCM to improve its robustness. The bias-corrected FCM (BCFCM) model was presented by Ahmed et al. [22], which introduced the Euclidean distance between pixels in the neighborhood and its cluster center as the constraint term for the objective function of fuzzy clustering. However, the spatial information used in BCFCM is not anisotropic (i.e., pixels in the neighborhood have same the weight for central pixel). Also, the distance between neighborhood and cluster center must be recalculated during each iteration. To decrease time consumption and complexity, Chen et al. [23] first filtered the image before iterative operation, and then proposed two improved models using mean and median filtering, namely, FCM_S1 and FCM_S2, respectively. Both models can reduce the energy minimization time. Nonetheless, both mean and median filtering inevitably blur the edge information. Further, the degree of fuzzy of segmentation results gradually rises with the increasing number of iteration steps. Klinids et al. [24] presented the fuzzy local information c-means (FLICM) model by introducing fuzzy parameters to directly manipulate the image and to reduce the number of input parameters. FLICM is highly robust, but it easily obtains incorrect segmentation results at the boundaries of different tissues and does not take the bias field into consideration. Ji et al. [25] incorporated nonlocal and local information into the energy formulation and designed a weighted image patch-based FCM (WIPFCM) model to further improve robustness. They substituted the original image with image patches, which provided anisotropic weighting for the image pixels.

Most of the models described above cannot correct the bias field caused by intensity inhomogeneity. Hence, Li et al. designed a local intensity clustering (LIC) [26] and multiplicative intrinsic component optimization (MICO) model [27], respectively. The two models apply an edge-based level set model and a clustering model, respectively. MICO also takes full advantage of a smoothly and slowly changing bias field. Both LIC as well as MICO can correct the bias field of an inhomogeneous intensity image, but they do not deal well with noise in MRI. Another popular model is the bias correction embedded fuzzy c-means (BCEFCM) [28] proposed by Feng et al., which adds the compute unified device architecture (CUDA) accelerated nonlocal means denoising model to remove noisy pixels. They assumed an original image consisting of a real image and a bias field, and then convolved the bias field by a kernel function in each iteration.

To address the problem of brain MRI being destroyed by noise and intensity inhomogeneity, a novel robust clustering with local contextual information (RC_LCI) model is presented here for simultaneously segmenting brain MRI and correcting the bias field. First, a weighting scheme is constructed to remove noisy pixels and thus update each pixel in the original image, which can improve robustness to noise. Second, a multiplicative framework is used to correct the bias field, in which the observed image is regarded as the product of the real image and the bias field. It is noteworthy that noisy pixels have been almost removed by the weighting scheme. Also, the optimal linear combination of basis functions can be applied to represent the bias field to guarantee that it is smoothly varying. When the energy function is minimized, the bias field and segmentation results can be obtained simultaneously.

The rest of this paper is organized as follows. Two state-of-the-art models are reviewed in Section 2. Section 3 describes in detail the proposed RC_LCI model. Section 4 shows numerous experimental results on synthetic and real brain MRI. Finally, the conclusions of this paper are given in Section 5.

## 2. Related Work

### 2.1. Bias-Corrected Fuzzy C-Means (BCFCM)

A brain MRI can be easily corrupted by noise, so an FCM model that does not consider spatial information cannot accurately segment such images. Ahmed et al. [22] modified FCM by introducing

a spatial constraint as the regularization term to allow central pixel $x_k$ to be determined by pixels $x_r$ in its immediate neighborhood. The objective function of BCFCM is given by

$$J_{BCFCM} = \sum_{i=1}^{c}\sum_{k=1}^{N} u_{ik}^{m}\|x_k - v_i\|^2 + \frac{\alpha}{N_R}\sum_{i=1}^{c}\sum_{k=1}^{N} u_{ik}^{m}\sum_{r\in N_k}\|x_r - v_i\|^2, \tag{1}$$

where $u_{ik}^{m}$ represents the membership function, $v_i$ is the cluster center, $N_k$ stands for the set of neighbors, and $\alpha$ is a parameter that needs to be manually adjusted to control the influence of the regularization term. The objective function can be minimized in a way that is similar to the standard FCM model, so the minimization of $u_{ik}$ can be expressed as

$$u_{ik} = \frac{\left(\|x_k - v_i\|^2 + \frac{\alpha}{N_R}\sum_{r\in N_k}\|x_r - v_i\|^2\right)^{-\frac{1}{(m-1)}}}{\sum_{j=1}^{c}\left(\|x_k - v_j\|^2 + \frac{\alpha}{N_R}\sum_{r\in N_k}\|x_r - v_j\|^2\right)^{-\frac{1}{(m-1)}}}. \tag{2}$$

Then, the minimization of $v_i$ can be expressed as

$$v_i = \frac{\sum_{k=1}^{N} u_{ik}^{m}\left(x_k + \frac{\alpha}{N_R}\sum_{r\in N_k} x_r\right)}{(1+\alpha)\sum_{k=1}^{N} u_{ik}^{m}}. \tag{3}$$

Though BCFCM integrates spatial information, noise pixels still destroy the central pixel because of the same weight for each pixel in the immediate neighborhood. Further, the regularization term needs to be computed during each iteration, which takes much more time than other models.

## 2.2. Fuzzy Local Information C-Means (FLICM)

The FLICM [24] model was presented by Krinidis et al. FLICM does not need to manually input a number of parameters but directly introduces fuzzy parameters to the objective function. The expression of the fuzzy parameter is given by

$$G_{ki} = \sum_{j\in N_k, j\neq k}\frac{1}{d_{kj}+1}\left(1 - u_{ij}\right)^{p}\|x_j - v_i\|^2, \tag{4}$$

where $x_j$ represents the pixel within the neighborhood, $v_i$ represents the cluster center, $u_{ij}$ represents the membership matrix, and $d_{kj}$ represents the Euclidean distance between the $j$-th and $k$-th pixels. After introducing fuzzy parameters, the objective function of FLICM can be represented as

$$J_m = \sum_{i=1}^{N}\sum_{k=1}^{c}\left[u_{ki}^{m}\|x_i - v_k\|^2 + G_{ki}\right], \tag{5}$$

where $u_{ki}$ is deduced by

$$u_{ki} = \left(\sum_{j=1}^{c}\left(\frac{\|x_i - v_k\|^2 + G_{ki}}{\|x_i - v_j\|^2 + G_{ji}}\right)^{\frac{1}{m-1}}\right)^{-1}. \tag{6}$$

The expression of $v_k$ is given by

$$v_k = \frac{\sum\limits_{i=1}^{N} u_{ki}^m x_i}{\sum\limits_{i=1}^{N} u_{ki}^m}. \tag{7}$$

FLICM has strong robustness to noise, but it is not always effective in practical applications and incorrect segmentation may occur at the boundaries of different tissues. The reason for this is that the gray values of two adjacent tissues are greatly similar, so a wrong value of $G_{ki}$ will be obtained in both tissues.

## 3. Robust Clustering with Local Contextual Information (RC_LCI)

Some clustering models do not pay any attention to the local information of the image; hence, the results are often sensitive to noise and outliers. Thus, the RC_LCI model is proposed in this section to segment inhomogeneous intensity brain MRI with noise. Brain MRI is composed of two parts: background and foreground, including gray matter (GM), white matter (WM), and cerebrospinal fluid (CSF). As shown in Figure 1, the RC_LCI model consists of two main parts: an anisotropic weighting scheme and a bias correction model. From the anisotropic weighting scheme in the first and second columns of Figure 1, the original image is destroyed by severe noise and intensity inhomogeneity (Figure 1a) in the foreground. Thus, the corresponding histogram of the original image does not have well-defined peaks. In contrast, the noise-removed image (Figure 1b) is more noise free than the original image, but it still has no well-separated peaks in the corresponding histogram owing to intensity inhomogeneity. Then, the bias field correction model in the third and fourth columns of Figure 1 can effectively correct bias field. More details about the RC_LCI model are described below.

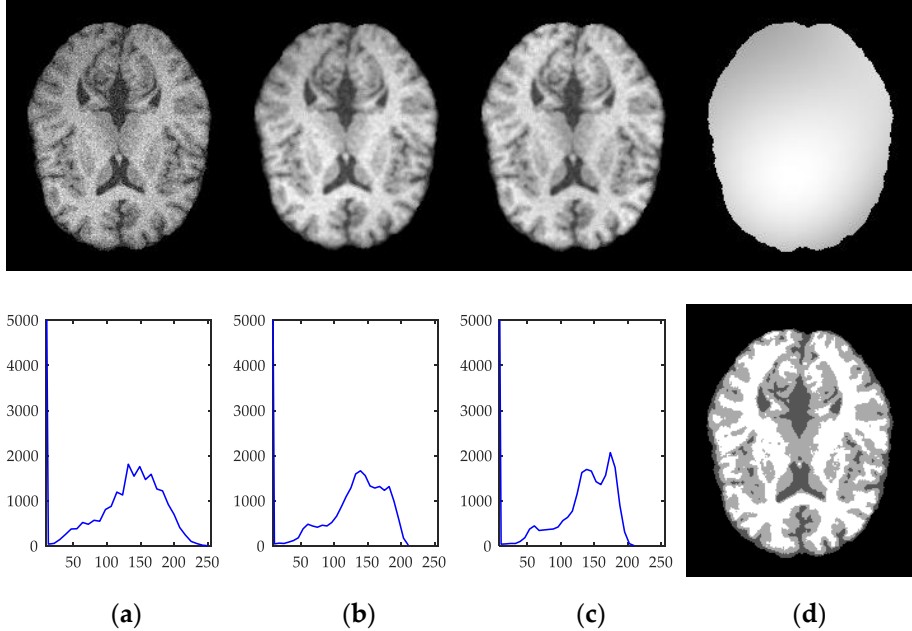

**Figure 1.** The composition of the robust clustering with local contextual information (RC_LCI) model: (**a**) original image and corresponding histogram; (**b**) noise-removed image and corresponding histogram; (**c**) bias field correction image and corresponding histogram; and (**d**) estimated bias field and segmentation result.

### 3.1. Anisotropic Weighting Scheme

In the anisotropic weighting scheme, each pixel in the neighborhood has a different effect on its central pixel $x$. That is, a pixel similar to $x$ has a larger weight and vice versa. Thus, the weight of each

pixel in the neighborhood and the corresponding gray value can be used to update the current central pixel and then generate a new central pixel value.

Assuming that a pixel within its 8-neighborhood $N_x$ is $y$, then $y \in N_x$. The gray mean square error between pixel $y$ and another pixel $y'$ in the 8-neighborhood is first calculated to obtain the difference of the gray value for all pixels in the 8-neighborhood:

$$\sigma_{xy} = \left[ \frac{\sum_{y' \in N_x \backslash \{y\}} (I_{y'} - I_y)^2}{n_x - 1} \right]^{\frac{1}{2}}, \tag{8}$$

where $I_y$ and $I_{y'}$ represent the corresponding gray value, and $n_x$ shows the number of pixels within the neighborhood $N_x$ (i.e., $n_x = 8$). $\sigma_{xy}$ can be used to generate the weight of each pixel in the neighborhood:

$$\gamma_{xy} = \exp \left[ -\left( \sigma_{xy} - \frac{\sum_{y \in N_x} \sigma_{xy}}{n_x} \right) \right]. \tag{9}$$

Finally, these weights are normalized as

$$\eta_{xy} = \frac{\gamma_{xy}}{\sum_{y \in N_x} \gamma_{xy}}. \tag{10}$$

Therefore, $x$ is updated according to similar pixels in the neighborhood and corresponding weights:

$$I_w(x) = \sum_{y \in N_x} I_y \times \eta_{xy}. \tag{11}$$

The gray value $I_w(x)$ can be used to update all $x$ in the original image.

### 3.2. Bias Field Framework

The observed MRI $I$ can be regarded as a multiplicative framework composed of the product of the bias field and the real image to more accurately segment the image and estimate the bias field:

$$I(x) = b(x)J(x) + n(x), \tag{12}$$

where $b(x)$ represents the bias field. $J(x)$ is the real image that reflects the physical properties of different tissues; so under ideal conditions, pixel $x$ in the uniform tissue should take the same value. $n(x)$ is the zero-mean additive Gaussian noise. According to the smoothly varying property of the bias field, $b(x)$ is defined by

$$b(x) = \sum_{i=1}^{M} w_i g_i, \tag{13}$$

where $g_1, \cdots, g_M$ are the basis functions and $w_1, \cdots, w_M$ are the optimal coefficients. In this study, the linear combination of multiple-order Legendre polynomial functions was used as the basis functions to ensure the property of the bias field. They can express as a column vector

$$G(x) = (g_1(x), \cdots, g_M(x))^T, \tag{14}$$

where $(\cdot)^T$ is the transpose operator. Then, the optimal coefficients $w_1, \cdots, w_M$ also can be represented by a column vector $\mathbf{w} = (w_1, \cdots, w_M)^T$. Thus, $b(x)$ is re-represented as

$$b(x) = \mathbf{w}^T G(x). \tag{15}$$

Image domain $\Omega$ is divided into $N$ different tissues, where $\Omega_i$ represents the $i$-th tissue. The $\Omega_1, \cdots, \Omega_N$ form each partition of $\Omega$ (i.e., $U_{i=1}^{N} \Omega_i = \Omega$ and $\Omega_i \cap \Omega_j = \varnothing$). In the $i$-th tissue, real image $J(x)$ is approximated as a constant $c_i$ and each $\Omega_i$ is represented by its membership function $u_i$. Ideally,

each pixel is contained by only one tissue, so $u_i$ is a binary function (i.e., $x \in \Omega_i, u_i(x) = 1$ or $x \notin \Omega_i, u_i(x) = 0$). Thus, real image $J(x)$ is given by

$$J(x) = \sum_{i=1}^{N} c_i u_i(x). \tag{16}$$

### 3.3. Energy Formulation

In this subsection, the energy function $F$ is constructed on the basis of the expression of the noise-removed image, bias field, and real image. A standard K-means algorithm can be used to classify pixels in the entire image domain and, thus, $F$ is defined by

$$F(b, J) = \int_{\Omega} |I_w(x) - b(x)J(x)|^2 dx, \tag{17}$$

where $I_w(x)$ is the denoising image after using the anisotropic weighting scheme. The expressions of $b(x)$ and $J(x)$ described in Equations (15) and (16) can be introduced to minimize this energy function effectively. Then, the energy function can be re-expressed as an equation with respect to three independent variables $\mathbf{w} = (w_1, \cdots, w_N)^T$, $\mathbf{c} = (c_1, \cdots, c_N)^T$, and $\mathbf{u} = (u_1, \cdots, u_N)^T$; so, $F$ is given by

$$F(b, J) = F(\mathbf{w}, \mathbf{c}, \mathbf{u}) = \int_{\Omega} \lambda_i \left| I_w(x) - \mathbf{w}^T G(x) \sum_{i=1}^{N} c_i u_i(x) \right|^2 dx, \tag{18}$$

where $\lambda_i$ is the weighting coefficient for the $i$-th cluster center. Since exchanging the order of the integrals and summations does not change the equation and $u_i$ is a binary membership function in each $\Omega_i$, the energy equation can be re-presented as

$$F(\mathbf{w}, \mathbf{c}, \mathbf{u}) = \int_{\Omega} \lambda_i \sum_{i=1}^{N} \left| I_w(x) - \mathbf{w}^T G(x) c_i \right|^2 u_i(x) dx. \tag{19}$$

### 3.4. Energy Minimization

The optimization of $b(x)$ and $J(x)$ can be obtained by minimizing the energy function $F$ about three independent variables $\mathbf{w}$, $\mathbf{c}$, and $\mathbf{u}$. The process of energy minimization is achieved through alternative iterations, where variables $\mathbf{w}$, $\mathbf{c}$, and $\mathbf{u}$ are obtained by fixing the other two variables in the last iteration. By fixing $\mathbf{c}$ and $\mathbf{u}$, the optimal solution of $\mathbf{w} = (w_1, \cdots, w_N)^T$ can be obtained by solving the partial derivative

$$\frac{\partial F}{\partial \mathbf{w}} = -2 \int_{\Omega} G(x) I_w(x) \left( \lambda_i \sum_{i=1}^{N} c_i u_i(x) \right) dx + 2\mathbf{w} \int_{\Omega} G(x) G^T(x) \left( \lambda_i \sum_{i=1}^{N} c_i^2 u_i(x) \right) dx. \tag{20}$$

Let Equation (20) equal 0, $\mathbf{w}$ is given by

$$\mathbf{w} = \left( \int_{\Omega} G(x) G^T(x) \left( \lambda_i \sum_{i=1}^{N} c_i^2 u_i(x) \right) dx \right)^{-1} \int_{\Omega} G(x) I_w(x) \left( \lambda_i \sum_{i=1}^{N} c_i u_i(x) \right) dx. \tag{21}$$

Hence, bias field $b(x)$ is obtained from the product of the basis functions and the optimal solution of $\mathbf{w}$. By fixing $\mathbf{w}$ and $\mathbf{u}$, the optimal solution of $\mathbf{c} = (c_1, \cdots, c_N)^T$ can be obtained by

$$c_i = \frac{\int_{\Omega} I_w(x) b(x) u_i(x) dx}{\int_{\Omega} b^2(x) u_i(x) dx}, \quad i = 1, \cdots, N. \tag{22}$$

The binary function represents the membership function of each tissue, which can ensure that noisy free pixels can be correctly classified. So, the function $\mathbf{u} = (u_1, \cdots, u_N)^T$ can be expressed as

$$u_i(x) = \begin{cases} 1, i = \arg\min_{i}\{\alpha_i[I_w(x)]\} \\ 0, i \neq \arg\min_{i}\{\alpha_i[I_w(x)]\} \end{cases}, \tag{23}$$

where $\alpha_i(x) = \left|I_w(x) - \mathbf{w}^T G(x)c_i\right|^2$.

In this study, 10 orthogonal three-order Legendre polynomial functions were used as the basis functions of bias field $b(x)$, which can be represented by $g_1(x) = 1$, $g_2(x) = x_1$, $g_3(x) = (3x_1^2 - 1)/2$, $g_4(x) = (5x_1^3 - 3x_1)/2$, $g_5(x) = x_2$, $g_6(x) = x_1x_2$, $g_7(x) = (3x_1^2 - 1)x_2/2$, $g_8(x) = (3x_1^2 - 1)/2$, $g_9(x) = (3x_2^2 - 1)x_1/2$, and $g_{10}(x) = (5x_2^3 - 3x_2)/2$, where $x_1$ and $x_2$ are directional components of image $I(x)$, and $\lambda_i = 1, i = 1, \ldots, N$. Then, the specific implementation process of RC_LCI can be summarized as the pseudocode in Algorithm 1.

---

**Algorithm 1.** RC_LCI segmentation algorithm.

---

1. **Input**: Brain MRI to be segmented.
2. Image = denoising(Image, $n_k$)          % Update each pixel according to Equations (8)–(11).
3. Basis = GetBasisOrder3(size(Image))       % Obtain basis functions according to Equation (14).
4. Randomly initialize bias field $b$, cluster center $C$, and membership function $M$.
5.     **while** $C^{(n)} - C^{(n-1)} > 0.001$ **do**
6.             **for** $i = 1 : \text{size(Basis, 3)}$
7.                   $v(i) = \text{Basis} * \text{Image} * C * M$
8.                   **for** $j = i : \text{size(Basis, 3)}$
9.                         $A(i, j) = \text{Basis} * C^2 * M$
10.                        $A(j, i) = A(i, j)$
11.                 **end for**
12.           **end for**
13.   $w = inv(A) * v$
14.           **for** $i = 1 : \text{size(Basis, 3)}$
15.                 $b = b + w(k) * \text{Basis}(k)$
16.           **end for**
17.           **for** $n = 1 : \text{size}(M, 3)$
18.                 $N = b * \text{Image} * M$
19.                 $D = b^2 * M$
20.                 $C(n) = N / D$
21.                 $e(i) = (\text{Image} - C(i) * b)^2$
22.           **end for**
23.   N_min = min($e$, [[], 3)
24.           **for** $k = 1 : \text{size}(e, 3)$
25.                 $M(:, :, k) = (\text{N\_min} == k)$
26.           **end for**
27.   **end while**
28.   New_Image = $C * M$
29.   Image_bc = New_Image ./ $b$            % Bias field correction image
30. **Output:** Bias field $b$, bias field correction image, and segmentation result.

---

## 4. Experimental Results

In order to demonstrate different aspects of the performance of RC_LCI, it was first applied to segment images with noise. Then, it was applied to correct inhomogeneous intensity images with different angles. Finally, images with noise and intensity inhomogeneity were used to demonstrate

the effectiveness of RC_LCI. Experimental results were tested in Matlab R2015b on a computer with Core(TM) i7-8700, 8 GB of RAM, and Windows 10 operating system.

### 4.1. Robustness to Noise

Both synthetic and real brain MRI were used to show the robustness of RC_LCI. BrainWeb [29] is a simulated brain database used to simulate brain MRI using three sequences (T1-, T2-, and PD-weighted) and a variety of slice thicknesses, noise levels, and levels of intensity inhomogeneity; the brain images used in this paper were T1-weighted and had a 1-mm slice thickness. Firstly, synthetic images with 9% noise provided by BrainWeb were used as the experimental images and RC_LCI was compared with MICO [27], BCFCM [22], and FLICM [24]. In Figure 2, the upper row shows the 67th slice image and the lower row shows the 96th slice image. It can be seen from the original images in the first column that WM and GM were intertwined with each other due to severe noise, which meant that it was difficult to segment them accurately without an effective model. Since MICO does not consider local information and thus is sensitive to noisy pixels, it produces incorrect results in the case of severe noise. Therefore, it can be clearly observed in the second column that there were many outliers in the segmentation results of MICO. The third column shows that noisy pixels were clearly presented in the segmentation results of BCFCM because the local spatial information was not anisotropic, so the noisy pixels in the neighborhood directly led to misclassification. FLICM utilizes the spatial Euclidean distance in the neighborhood, which easily results in oversegmentation in adjacent different areas. RC_LCI utilizes the weight of each pixel in the 8-neighborhoods to update the current central pixel, so it can easily identify the noisy pixels in the neighborhood so as to decrease the influence of noise. By comparing the segmentation results of all models with the ground truth (GT) in the sixth column, it can be seen that the segmentation results of RC_LCI were the closest to GT. From the above experimental results, the robustness of RC_LCI to noise was slightly superior to FLICM and significantly superior to MICO and BCFCM. The iterations and Central Processing Unit (CPU) time of the four models are listed in Table 1. It can be seen from the last row that RC_LCI consumed less iterations and CPU time than the other models in terms of segmentation efficiency.

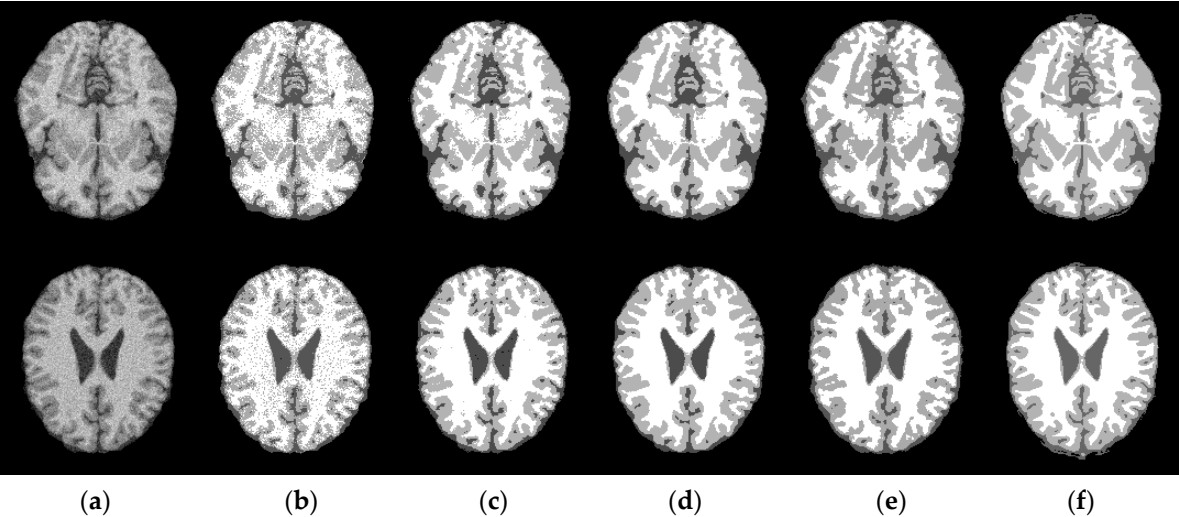

(**a**)　　　　(**b**)　　　　(**c**)　　　　(**d**)　　　　(**e**)　　　　(**f**)

**Figure 2.** Comparison of the segmentation results of multiplicative intrinsic component optimization (MICO), bias-corrected fuzzy c-means (BCFCM), fuzzy local information c-means (FLICM), and RC_LCI: (**a**) original images; (**b**) segmentation results of MICO; (**c**) segmentation results of BCFCM; (**d**) segmentation results of FLICM; (**e**) segmentation results of RC_LCI; and (**f**) ground truth.

**Table 1.** The iterations and CPU time of MICO, BCFCM, FLICM, and RC_LCI.

| Model | 67th Slice | | 96th Slice | |
|---|---|---|---|---|
| | Iteration | Time (s) | Iteration | Time (s) |
| MICO | 9 | 4.79 | 8 | 3.97 |
| BCFCM | 26 | 25.77 | 23 | 22.56 |
| FLICM | 44 | 6.61 | 40 | 5.83 |
| RC_LCI | 11 | 3.83 | 9 | 3.16 |

RC_LCI was then applied to real images with skulls and compared with MICO, BCFCM, and FLICM. In Figure 3, axial-, sagittal-, and coronal-sectioned brain MRI are shown in the first, second, and third rows, respectively. As shown in Figure 3, GM and WM are also intertwined with each other in the first column and the three images are occupied by low noise. The segmentation results of MICO were still corrupted by noisy pixels. BCFCM did not clearly distinguish WM and GM in adjacent areas. FLICM did not distinguish background and CFS because of oversegmentation, so it obtained incorrect segmentation results when the background was adjacent to CFS. In real brain MRI, RC_LCI clearly classified images into four classes and effectively removed noisy pixels without being influenced by the skull.

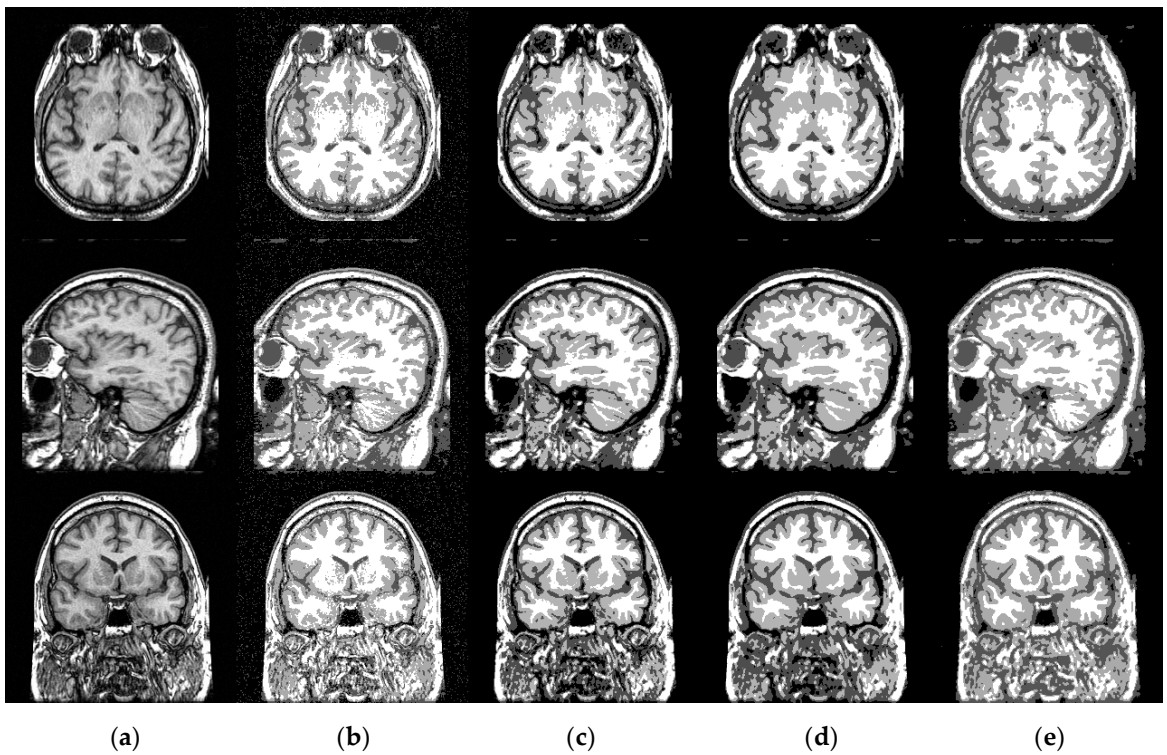

(a)  (b)  (c)  (d)  (e)

**Figure 3.** The segmentation results of RC_LCI compared with MICO, BCFCM, and FLICM on real images: (**a**) original real brain MRI; (**b**) segmentation results of MICO; (**c**) segmentation results of BCFCM; (**d**) segmentation results of FLICM; and (**e**) segmentation results of RC_LCI.

To more objectively and accurately compare the performances of these models, quantitative evaluation was used to analyze the segmentation results of synthetic brain images with noise ranging from 6% to 9%. BrainWeb also provides GT, which are the standard segmentation results of the corresponding images. The Jaccard similarity (JS) coefficient can be used to compare the similarity of two sets. It is given by

$$JS(S_1, S_2) = \frac{|S_1 \cap S_2|}{|S_1 \cup S_2|},$$

(24)

where $S_1$ and $S_2$ are two segmentation results. $|\cdot|$ shows the number of pixels in the corresponding area. The more accurate the segmentation results, the higher the JS. The JS coefficient for GM, WM, and CSF acquired from MICO, BCFCM, FLICM, and RC_LCI are shown in the boxplots of Figure 4. As shown in Figure 4, RC_LCI had the highest JS compared with the other three models and had no outliers in its JS, which revealed that RC_LCI not only had better robustness to noise but also higher accuracy than the other three models. Both visual and quantitative evaluations showed that the RC_LCI model was more robust to different noises and could obtain more accurate results than the other models, whether considering synthetic or real images.

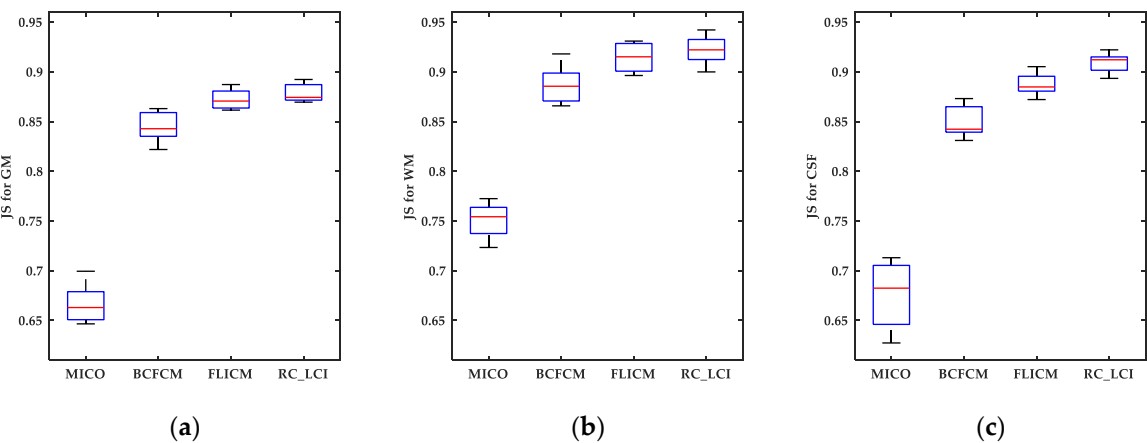

**Figure 4.** Jaccard similarity coefficients of MICO, BCFCM, FLICM, and RC_LCI: (**a**) Jaccard similarity (JS) for gray matter (GM); (**b**) JS for white matter (WM); and (**c**) JS for cerebrospinal fluid (CSF).

### 4.2. Capability of Estimating the Bias Field

Synthetic inhomogeneous intensity brain MRI with different angles obtained from BrainWeb were used to show the capability of RC_LCI in estimating the bias field. Axial-, sagittal-, and coronal-sectioned images are shown in the first, second, and third rows of Figure 5, respectively. As shown in Figure 5, the estimated bias field in the second column satisfied the smoothly varying property due to the orthogonal basis functions. Hence, bias field correction images were more homogeneous than the original images, which can be seen from the third column. As shown in the last column, GM, WM, CSF, and background were distinctly segmented into four classes.

The histograms of the original and corresponding bias field correction images obtained from Figure 5 are shown in the first and second rows of Figure 6, respectively. The histograms of the original and bias field correction images can be contrasted to further prove that the quality of the image was effectively improved. The first row shows that the intensity of GM, WM, and CSF could not be clearly distinguished, which meant that these images could not be segmented into four classes accurately. There was no obvious peak that represented each tissue in the histogram of the original inhomogeneous intensity image. On the contrary, the well-defined peak can be seen in the histogram of the bias field correction image. The above comparison demonstrates that RC_LCI can obtain accurate segmentation results from inhomogeneous intensity brain MRI.

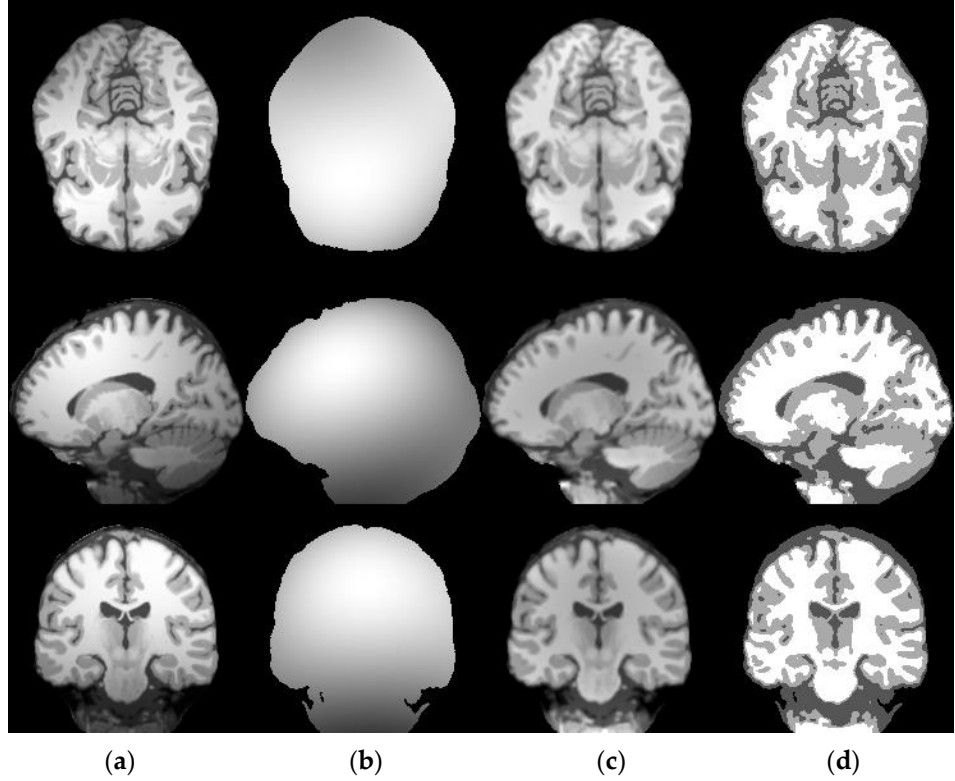

**Figure 5.** Segmentation results of inhomogeneous intensity brain MRI: (**a**) original images; (**b**) estimated bias field; (**c**) bias field correction images; and (**d**) segmentation results.

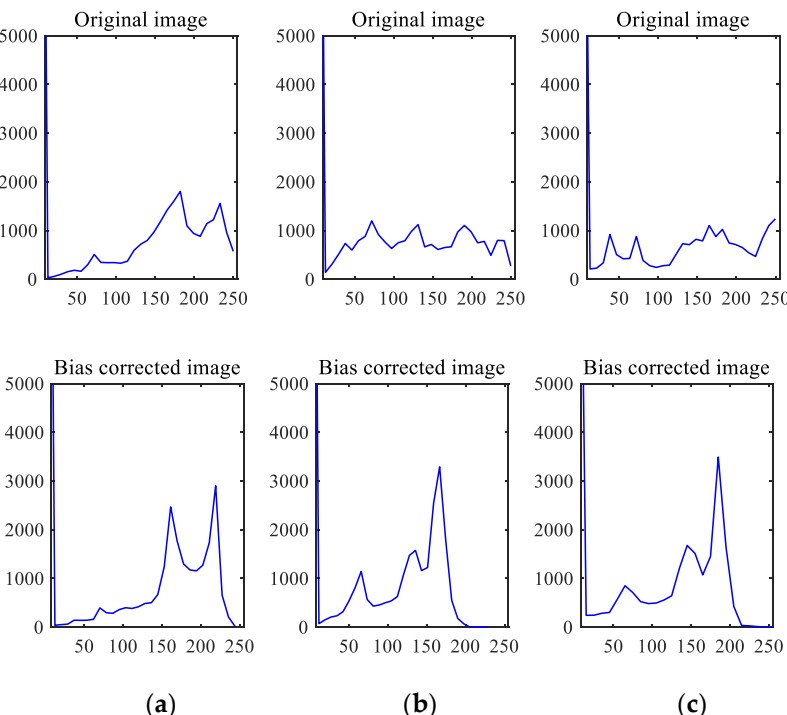

**Figure 6.** The histograms of the original and bias field correction images: (**a**) the histogram of axial-sectioned images; (**b**) the histogram of sagittal-sectioned images; and (**c**) the histogram of coronal-sectioned images.

### 4.3. Effectiveness of RC_LCI

Synthetic brain MRI with 7% noise and 80% intensity inhomogeneity obtained from BrainWeb were used to demonstrate the effectiveness of RC_LCI, and the results of the comparison with two state-of-the-art models are shown in Figure 7. The segmentation results of MICO, LIC [26], and RC_LCI are shown in the first to third rows, respectively. MICO could not effectively remove noisy pixels from the original image. The bias field estimated by LIC could not ensure its slowly varying property. RC_LCI used three-order orthogonal basis functions as the basis functions, which not only ensured the property of the bias field but also obtained accurate segmentation results. In addition, it can be intuitively observed that the boundaries between different tissues could be clearly distinguished.

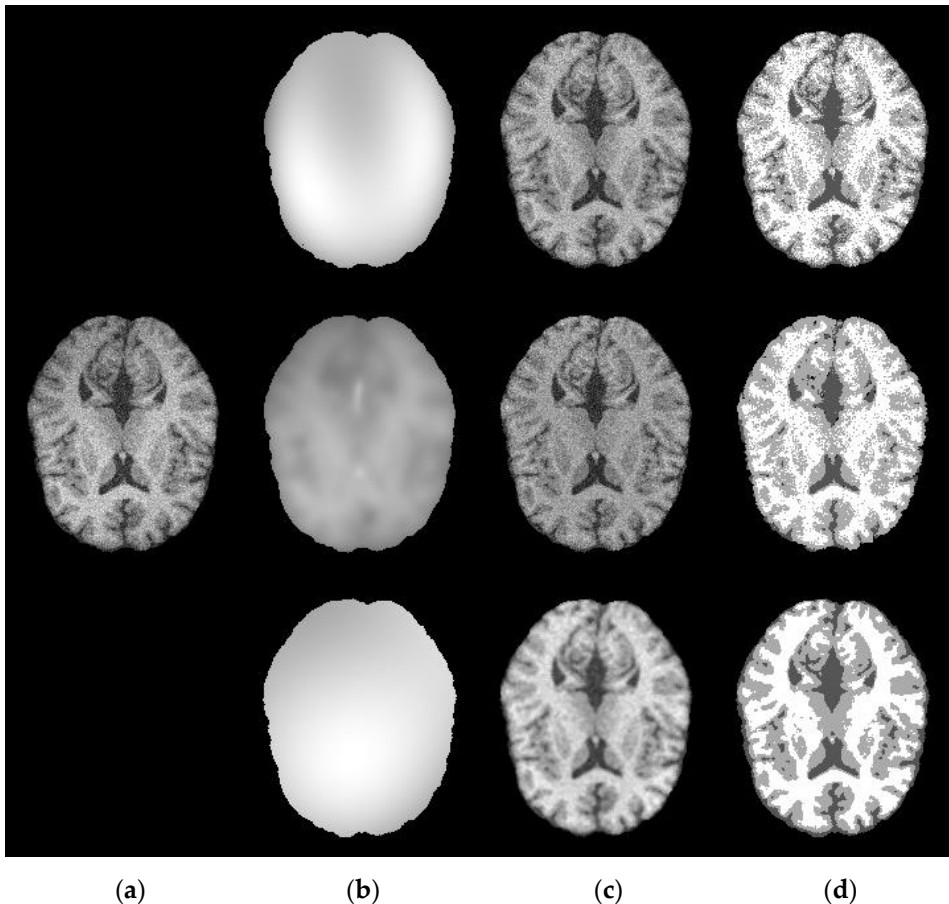

(**a**)　　　　　　　　(**b**)　　　　　　　　(**c**)　　　　　　　　(**d**)

**Figure 7.** Comparison of the segmentation results of local intensity clustering (LIC), MICO, and RC_LCI: (**a**) original image; (**b**) estimated bias field; (**c**) bias field correction images; and (**d**) segmentation results.

The visual results cannot fully demonstrate the performance of RC_LCI. Thus, the JS coefficient was used to quantitatively evaluate the segmentation results and show the discrepancies between the segmentation results of each model and GT. It is noteworthy that the better match between results and GT will lead to higher values for JS. The JS of GM, WM, and CSF obtained from LIC, MICO, and RC_LCI in Figure 7 are displayed in Table 2. The JS coefficients of RC_LCI were markedly superior to those obtained by LIC and MICO, which reveals that RC_LCI can effectively segment brain MRI with both noise and intensity inhomogeneity. Therefore, both visual results and objective evaluations can demonstrate the effectiveness of RC_LCI.

**Table 2.** JS of LIC, MICO, and RC_LCI.

| Tissues | LIC | MICO | RC_LCI |
|---------|-------|-------|--------|
| GM | 0.768 | 0.735 | 0.842 |
| WM | 0.815 | 0.783 | 0.889 |
| CSF | 0.602 | 0.691 | 0.803 |

## 5. Conclusions

A novel robust clustering with local contextual information model was proposed for segmenting brain MRI with noise and intensity inhomogeneity. An anisotropic weighting scheme was introduced to improve robustness, which could make full use of local spatial information and update the current central pixel according to pixels in the immediate neighborhood. Then, a multiplicative framework consisting of the product of the real image and the bias field was used to correct the bias field and segment brain MRI. A linear combination of three-order Legendre polynomial functions was used as the basis functions of the bias field to guarantee its property. Finally, energy formulation was defined based on the expression of the weighting scheme, bias field, and real image. Visual experimental results and objective evaluations have proved that RC_LCI can more effectively remove noisy pixels and obtain accurate segmentation results than the compared models for real and synthetic images. As a result, the segmentation results obtained by RC_LCI were increased by $0.195 \pm 0.125$ in terms of JS. Future research will consider the level set model or global information to deal with complex brain MRI.

**Data Availability:** The data and code used to support the findings of this study are available from the corresponding author upon request.

**Author Contributions:** Software, Z.Z.; validation, J.S.; methodology, J.S. and Z.Z.; resources, J.S.; data curation, Z.Z. and J.S.; writing—original draft preparation, Z.Z.; writing—review and editing, J.S.; funding acquisition, J.S.

**Funding:** This research was funded by the Principal Fund Project of Minnan Normal University (Grant No. KJ18010), the Education and Teaching Reform Project of Undergraduate Colleges and Universities in Fujian Province (Grand No. FBJG20180015) and the Fujian Provincial Natural Science Foundation Project (Grant No. 2017J01708).

**Conflicts of Interest:** The authors declare no conflict of interests.

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
