# Peer review of "A Robust Brain MRI Segmentation and Bias Field Correction Method Integrating Local Contextual Information into a Clustering Model"

_applsci, doi:10.3390/app9071332_

Round 1

Reviewer 1 Report

Overall a high quality paper showing a small but important increase in segmentation quality. Some minor English errors but reads well. Good results, not always a clearly presented as possible.

Minor comments:

The acronym MRI for magnetic resonance imaging is more conventional - I believe the authors should use this throughout as it will optimise their paper for academics searching the literature and make it more accessible to less-specialist readers.

Many minor English issues throughout, e.g. 'error segmentation' at the end of paragraph one (Line 34) should read 'segmentation error'. Please get this proof read.

Line 35: What is a 'ripe' method?

Line 59 is unintelligible.

Figures are not near to their references in the text - this may be a template issue but it should be addressed before publication.

Please rearrange the table/figure columns/plots in the order: MICO, BCFCM, FLICM and RC_LCI. Given the latter two give the most similar results they should be near each other for easy comparison.

Conclusions for Figure 2 (lines 225-239) are difficult to agree with based on the images shown. Can a ground truth example be provided?

Please provide all settings used in BrainWeb, not just the noise level or intensity inhomogeneity.

Figure 4's caption is not sufficient. Also, your y axis is inconsistent.

The Jaccard Coefficient is used in section 4.1 but the Dice Coefficient in 4.3. Why? Is the Jaccard not suitable for both?

Major comments:

BCEFCM is described in detail in section 2 but BCFCM and not BCEFCM is used in the results? Please add BCEFCM to your experiments and BCFCM to section 2 (or more detail in section 1).

Why is FLICM not used in section 4.3 onwards? Please add this.

No information is given about access to this new algorithm. Please add this.

No information is given about implementation or how long this method takes to run on an average image (compared to other clustering methods). Please add this.

Author Response

Thank you very much for your valuable comments and the opportunity to revise our manuscript. We appreciate reviewer very much for these positive and constructive comments and suggestions for our manuscript. We have studied the comments of reviewer carefully and have revised our manuscript according to the comments. The corrections in the manuscript and the responds to the comments of reviewer are as following:

Minor comments:

Point 1: The acronym MRI for magnetic resonance imaging is more conventional - I believe the authors should use this throughout as it will optimise their paper for academics searching the literature and make it more accessible to less-specialist readers.

Response 1: It is really true as reviewer suggested that we should use acronym MRI for magnetic resonance imaging. Thus, we have made correction according to the suggestion of reviewer and replaced ‘MR image’ with ‘MRI’, which can make our paper more convenient to be read and searched by readers.

Point 2: Many minor English issues throughout, e.g. 'error segmentation' at the end of paragraph one (Line 34) should read 'segmentation error'. Please get this proof read.

Response 2: Thank you for your careful reading. We examined this sentence carefully and found that it was indeed our negligence. We have replaced ‘error segmentation’ with ‘segmentation error’. Besides, we have carefully revised the grammar of the full manuscript.

Point 3: Line 35: What is a 'ripe' method?

Response 3: What we want to express is that clustering model is often used to segment medical images, so it is not appropriate for ‘ripe’ to be used in this sentence. Accordingly, we think that ‘widely’ is more suitable for expressing our meaning than ‘ripe’, and we have made the corresponding correction in line 37. Line 37: Among all the segmentation methods, clustering model is widely used, where hard clustering and soft clustering are two main methods.

Point 4: Line 59 is unintelligible.

Response 4: We have re-written this sentence carefully according to the suggestion of reviewer. Lines 63-64: Most of the models described above cannot correct bias field caused by intensity inhomogeneity.

Point 5: Figures are not near to their references in the text - this may be a template issue but it should be addressed before publication.

Response 5: We have rearranged all the Figures and Tables according to the suggestion of reviewer, and placed them at the top or bottom of the corresponding page.

Point 6: Please rearrange the table/figure columns/plots in the order: MICO, BCFCM, FLICM and RC_LCI. Given the latter two give the most similar results they should be near each other for easy comparison.

Response 6: According to the suggestion of reviewer, we have rearranged the order of these models in Figures 2, 3, and 4, which is more convenient for readers to compare the segmentation accuracy of RC_LCI and other models.

Point 7: Conclusions for Figure 2 (lines 225-239) are difficult to agree with based on the images shown. Can a ground truth example be provided?

Response 7: In response to the request of reviewer, we have added the ground truth (GT) of the original image to the sixth column of Figure 2 so that the segmentation results can be more intuitively compared.

Point 8: Please provide all settings used in BrainWeb, not just the noise level or intensity inhomogeneity.

Response 8: Thanks to the suggestion of reviewer, we have added more details with respect to BrainWeb in lines 205-208. Lines 205-208: BrainWeb [29] is a simulated brain database used to simulate brain MRI using three sequences (T1-, T2-, and PD-weighted) and a variety of slice thicknesses, noise levels, and levels of intensity inhomogeneity, where the brain images used in this paper is T1-weighted and 1 mm slice thickness.

Point 9: Figure 4's caption is not sufficient. Also, your y axis is inconsistent.

Response 9: According to the comment of reviewer, we have supplemented the caption of Figure 4 and normalized the y axis. Besides, we also rearranged the order of these models in Figure 4 according to the above point 6.

Point 10: The Jaccard Coefficient is used in section 4.1 but the Dice Coefficient in 4.3. Why? Is the Jaccard not suitable for both?

Response 10: The why Dice Coefficient is used in section 4.3 is because our original intention is to use different quantitative evaluations to fully demonstrate the superiority of the proposed RC_LCI model. Jaccard Coefficient can suit not only section 4.1 but also section 4.3. Thus, we have replaced Dice Coefficient in section 4.3 with Jaccard Coefficient to objectively show the performance of RC_LCI model, which can be seen from the Table 2.

Major comments:

Point 1: BCEFCM is described in detail in section 2 but BCFCM and not BCEFCM is used in the results? Please add BCEFCM to your experiments and BCFCM to section 2 (or more detail in section 1).

Response 1: It is our negligence and we are sorry about this. Firstly, Bias corrected FCM (BCFCM) and bias correction embedded FCM (BCEFCM) are two completely different models. We have re-written the section 2 and found that BCFCM should be described in detail in this section instead of BCEFCM, because both RC_LCI and BCFCM model consider the spatial neighborhood information. Secondly, denoising method and the method for ensuring the property of bias field are different between RC_LCI model and BCEFCM model, so we do not use the BCEFCM model as the compared model considering the discrepancy of RC_LCI and BCEFCM. Accordingly, we introduced BCEFCM briefly in section 1 and described BCFCM in detail in section 2.

Point 2: Why is FLICM not used in section 4.3 onwards? Please add this.

Response 2: FLICM has the strong robustness for the images with only noise, but is cannot estimate and correct bias field caused by intensity inhomogeneity. The gray values of two adjacent tissues are greatly similar in the case of images corrupted by noise and intensity inhomogeneity, so FLICM will obtain a wrong fuzzy parameter G and thus result in incorrect segmentation. On the other hand, the main role of section 4.3 is to demonstrate the effectiveness of models in the face of brain images corrupted by noise and intensity inhomogeneity simultaneously, so we reckon that FLICM is a bit unsuitable for being used in section 4.3.

Point 3: No information is given about access to this new algorithm. Please add this.

Response 3: Lines 332-333: As suggested by the reviewer, the data and code used to support the findings of this study are available from the corresponding author upon request, which can be used to study and improve the proposed RC_LCI model more conveniently.

Point 4: No information is given about implementation or how long this method takes to run on an average image (compared to other clustering methods). Please add this.

Response 4: We have made correction according to the comment of reviewer. We have added the iterations and CPU time of MICO, BCFCM, FLICM, and RC_LCI model in section 4.1, which can be seen from the Table 1.

Thank you again for your comments and valuable suggestions.

Reviewer 2 Report

The paper describes a method for segmenting MRI images of the brain. They compare their method to some existing methods and demonstrate an improvement over the state of the art.

The paper was written in a sloppy fashion. Equations should appear as part of the text and have appropriate punctuation. For example, eq 2 should be followed by a comma and eq 3 should be followed by a period. Also, the line after the equation does not need to be indented. 

All mathematics terms must be in the math font and that font should not change. For example, eq 12 does not use the font used in the other equations.

There are frequent grammar errors. One example can be found on lines 108-113. There is a run-on sentence and some poor wording. The sentence on 148 is not valid sentence. 199 contains dF/dw which is not needed as the equation follows this. Sentence on 246-247 is not valid.

The sudo code is very sloppy. More effort is needed to make is correct. This includes the white space which does not make any sense. 

Make sure figures are on pages that they are references. I would also suggest that they appear on top or bottom of the page, not in the middle. 

Author Response

Thank you so much for your thoughtful comments. These comments are all valuable and very helpful for improving the quality of our manuscript, as well as the important guiding significance to our researches. We have studied comments carefully and have made corresponding correction. The corrections in the manuscript and the responds to the comments of reviewer are as following:

Point 1: The paper was written in a sloppy fashion. Equations should appear as part of the text and have appropriate punctuation. For example, eq 2 should be followed by a comma and eq 3 should be followed by a period. Also, the line after the equation does not need to be indented.

Response 1: We are very sorry for our sloppy fashion. According to the guiding advice of reviewer and the template of Applied Sciences, we have made all the equations as part of the text using the appropriate punctuation.

Point 2: All mathematics terms must be in the math font and that font should not change. For example, eq 12 does not use the font used in the other equations.

Response 2: Thanks to the important comments of reviewer, we have changed all the fonts used in mathematics terms according to the formatting of mathematical components in the template of Applied Sciences.

Point 3: There are frequent grammar errors. One example can be found on lines 108-113. There is a run-on sentence and some poor wording. The sentence on 148 is not valid sentence. 199 contains dF/dw which is not needed as the equation follows this. Sentence on 246-247 is not valid.

Response 3: Thank you for your careful reading. It is really our negligence and we are very sorry for our incorrect writing. We have made correction according to the comment of reviewer. Besides, we have carefully revised the grammar of the full manuscript.

Point 4: The sudo code is very sloppy. More effort is needed to make is correct. This includes the white space which does not make any sense.

Response 4: According to the comment of reviewer, we have added more details on how to implement RC_LCI model in Algorithm 1.

Point 5: Make sure figures are on pages that they are references. I would also suggest that they appear on top or bottom of the page, not in the middle.

Response 5: We have rearranged all the Figures and Tables according to the suggestion of reviewer, and placed them at the top or bottom of the corresponding page.

Thank you again for your comments and hope to learn more from you.

Round 2

Reviewer 1 Report

The many changes made by the authors have significantly improved this paper making it a much stronger submission. I have no further specific comments to make.

Reviewer 2 Report

Still could use some editing of Algorithm 1. For example, lines 13, 19, and 20 should have the same white space as line 6 and 14. Comments should not spill over to next line without an additional %.